# Formulation and Evaluation of Alcohol-Free Hand Sanitizer Gels to Prevent the Spread of Infections during Pandemics

**DOI:** 10.3390/ijerph18126252

**Published:** 2021-06-09

**Authors:** Rayan Y. Booq, Abdullah A. Alshehri, Fahad A. Almughem, Nada M. Zaidan, Walaa S. Aburayan, Abrar A. Bakr, Sara H. Kabli, Hassa A. Alshaya, Mohammed S. Alsuabeyl, Essam J. Alyamani, Essam A. Tawfik

**Affiliations:** 1National Center for Biotechnology, Life Science and Environment Research Institute, King Abdulaziz City for Science and Technology (KACST), Riyadh 11442, Saudi Arabia; rbooq@kacst.edu.sa (R.Y.B.); aabakr@kacst.edu.sa (A.A.B.); eyamani@kacst.edu.sa (E.J.A.); 2National Center for Pharmaceutical Technology, Life Science and Environment Research Institute, King Abdulaziz City for Science and Technology (KACST), Riyadh 11442, Saudi Arabia; abdualshehri@kacst.edu.sa (A.A.A.); falmughem@kacst.edu.sa (F.A.A.); waburayan@kacst.edu.sa (W.S.A.); Skabli@kacst.edu.sa (S.H.K.); halshaya@kacst.edu.sa (H.A.A.); 3Center of Excellence for Biomedicine, Joint Centers of Excellence Program, King Abdulaziz City for Science and Technology (KACST), Riyadh 11442, Saudi Arabia; nzaidan@kacst.edu.sa; 4Life Science and Environment Research Institute, King Abdulaziz City for Science and Technology (KACST), 6086, Riyadh 11442, Saudi Arabia; malsubyl@kacst.edu.sa

**Keywords:** hand sanitizer, microbes, infections, essential oils, antimicrobial, alcohol-free, pandemics

## Abstract

Hand hygiene is an essential factor to prevent or minimize the spread of infections. The ability to prepare an alcohol-free hand sanitizer (AFHS) with antimicrobial properties is crucial, especially during pandemics, when there are high demands and a low supply chain for ethanol and isopropanol. The objective of this study was to prepare AFHS gels based on natural materials that contain essential oils (EOs) that would be effective against a broad spectrum of pathogens. The results showed that the organoleptic characteristics of all prepared hand sanitizer gels were considered acceptable. The pH of the formulations was slightly acidic (circa 3.9) owing to the presence of aloe vera in large proportions (90% *v*/*v*), which is known for its acidity. The spreadability for all tested formulations was in the acceptable range. The antimicrobial effectiveness test demonstrated that the prepared hand sanitizer gels had antimicrobial activities against different gram-positive and gram-negative bacteria and *Candida albicans* yeast. The highest antibacterial effect was observed with tea tree oil hand sanitizers, which lack activity against the yeast, while clove oil hand sanitizers showed effectiveness against all microorganisms, including *Candida albicans*. The lavender hand sanitizer exhibited the least antimicrobial efficiency. The acceptability study on 20 human volunteers showed that the hand sanitizer gel containing 1.25% (*v*/*v*) clove oil did not produce any signs of skin irritation. This study suggested that the prepared natural hand sanitizer gel with 1.25% (*v*/*v*) clove oil can be a potential alternative to commonly used alcohol-based hand sanitizers (ABHS).

## 1. Introduction

New infections, bacterial or viral, have often raised significant threats to public health across the globe. One of these hazardous pathogens is severe acute respiratory syndrome coronavirus 2 (SARS-CoV-2), which is renowned to cause coronavirus disease 2019 (COVID-19) that was declared a global pandemic by the World Health Organization (WHO) at the beginning of 2020 [1]. After its discovery in Wuhan in December 2019, more than 150 million confirmed cases have occurred worldwide by April 2021. The preventive protocols to cope with COVID-19 are just supportive in order to minimize the spread of this disease as the best approach. Frequent and reliable handwashing is one of the many approaches adopted to prevent the transmission of the virus.

Secondary bacterial or fungal infection can be considered as one of the most common and serious complications related to viral infections, especially in the elderly. Secondary infections could lead to the escalation of the clinical complications, increase the need for intensive care, and raise the rate of mortality [2]. A recent report on COVID-19 related co-infections showed that the mortality rate of 15.2% was observed for patients with pneumonia caused by antibiotic-resistant strains of *Staphylococcus aureus* (*S. aureus*) and *Klebsiella pneumoniae* (*K. pneumoniae*) [3]. Moreover, a study by Plantefèvere et al. reported that 28% of intensive care unit (ICU) patients with severe SARS-CoV-2 pneumonia are co-infected with bacteria [4]. The use of an effective hand sanitizer is considered as an essential alternative to handwashing, and is one of the current protocols to prevent the spread of viral infections and related secondary infections, hence decreasing the need for intensive care administration and antibiotics use.

Following the outbreak of COVID-19, alcohol-based hand sanitizers (ABHS) have become a common alternative to conventional handwashing in healthcare and neighborhood settings as a preventative tool, causing an increased alcohol demand [5]. Several hand sanitizers with different variations are available. It is essential to consider the types of hand sanitizers that function effectively against pathogens. ABHS recommended by the WHO are mostly composed of ethanol, isopropyl alcohols, or hydrogen peroxides in varying combinations, in which the ethanol or isopropyl alcohol concentration is mainly at a range of 60–95% [6]. This concentration range can be considered as the active bactericidal concentration range for most ABHS [7]. The demand for alcohol has increased due to the manufacturing of ABHS, which reduced the global alcohol supply chain massively. The WHO proposed two formulations for lower volume production of ABHS due to the COVID-19 pandemic demands for alcohol. The first formulation consists of 80% (*v*/*v*) ethanol, 0.125% (*v*/*v*) hydrogen peroxide, and 1.45% (*v*/*v*) glycerol, while the second formulation contains 75% (*v*/*v*) isopropyl alcohol, 0.125% (*v*/*v*) hydrogen peroxide, and 1.45% (*v*/*v*) glycerol [8]. This urged the need to find an alternative alcohol-free hand sanitizer (AFHS) with a comparable antimicrobial activity to overcome the risk in the alcohol supply chain.

In this study, AFHS formulations prepared from natural ingredients that include aloe vera, vitamin E, glycerin, and different essential oils (EOs) were evaluated. These ingredients are also widely available in the market, which make them easily accessible. The use of aloe vera gel as the hand sanitizer vehicle was due to its natural moisturizing and germ-retarding abilities, as well as the competence of inhibiting some bacterial strains [9]. Vitamin E and glycerin were used for their ability to slow down rancidity (i.e., oxidation or hydrolysis of fats and oils) and to moisturize the skin, respectively [10]. The primary active compounds of the AFHS gels are EOs, which have a wide range of antimicrobial activities [11]. The antimicrobial activity of EOs is reported to be due to their hydrophobic nature that facilitates the partition of active components in the lipid of the bacterial cell membrane and mitochondria, hence reducing cytoplasmic membrane integrity [12]. For instance, it was demonstrated that a mixture of clove oil and cinnamon is effective against several fungal, yeast, and bacterial species, such as *Aspergillus flavus, Debaryomyces hanseni,* and *S. aureus* [12]. The natural compounds used to prepare the AFHS are commonly used ingredients in cosmetic applications owing to the variety of their properties [13,14]. However, an optimized concentration of these compounds should be used in the preparation of AFHS, as the increased proportion could lead to dermal sensitivity and skin irritation, according to previously reported studies [15,16]. 

In addition, the preparation of hand sanitizer in the form of gel has several advantages over other forms of hand sanitizers, such as liquid (spray) or foam. The key desirable properties of gel formulation are the ability to create a protective layer on the site of application and the longer protection time on the skin in comparison to the other hand sanitizer forms. The retention time of the hand gel is higher than the liquid and foam hand sanitizers, and it has a preferable moisturizing feeling and adherence property on the applied skin [17]. Therefore, hand gel was considered as a suitable hand sanitizer form for the preparation of AFHS in this study.

These formulated AFHS gels are aimed to control the spread of co-infections during pandemics. Following the preparation of hand sanitizer gels, the characterization and the evaluation of all prepared formulations were carried out in terms of organoleptic properties, pH measurement, rheological behavior, and gel spreadability. A microbiological test zone of inhibition was performed against different bacterial strains and *Candida albicans (C. albicans*) yeast to examine the antimicrobial activity of AFHS gels. Finally, an acceptability test was conducted to assess the safety of the prepared hand gels by determining any side effects, such as skin irritation and skin redness, which may arise from their application on human skin.

## 2. Materials and Methods

### 2.1. Materials

Aloe vera (raw purified aloe; Pure Aloe Force^®^, Herbal Answers, Inc., Saratoga Springs, NY, USA), glycerin (USP vegetable; Heritage Store, Park City, UT, USA), vitamin E (31,500 mg; Rexall Sundown, Inc., Boca Raton, FL, USA), Eos: clove oil, lavender oil, and tea tree oil (100% pure; Now Foods, Bloomingdale, IL, USA), and three commercially available hand sanitizer gels with over 60% alcohol content: C1 (ethanol-based hand sanitizer), C2 (alcohol denat-based hand sanitizer), and C3 (ethanol/isopropanol-based hand sanitizer), were all obtained from a local supermarket. Mueller–Hinton broth was purchased from Scharlab, S.L (Barcelona, Spain). Distilled water was generated through Milli Q, Millipore (Billerica, MA, USA) and was used throughout the study.

### 2.2. Methods

#### 2.2.1. Preparation of Hand Sanitizer Gels 

Several natural materials were used to prepare the hand sanitizer gels in pertinent proportions, including aloe vera, glycerin, and vitamin E, in addition to EOs. Each formulation was prepared by dispersing glycerin (5% *v*/*v*) to aloe vera gel (90% *v*/*v*) in a 250 mL beaker and mixed with gentle stirring at ambient temperature. EOs (at 2.5% *v*/*v* or 1.25% *v*/*v*) were then added dropwise with constant stirring to avoid air bubble formation and to obtain uniform and homogenous gels, followed by adding vitamin E (0.05% *v*/*v*). The remainder of each formula was completed by distilled water. Control formulation was prepared using the same components of the prepared hand sanitizer gels, but with no addition of EOs. Table 1 shows the composition of all prepared hand sanitizer gels and the control gel with each ingredient’s concentration. 

#### 2.2.2. Physiochemical Characterization and Evaluation of Hand Sanitizer Gels

##### Organoleptic Test

The prepared samples were inspected visually to check the texture, odor, and color of the gels in semisolid conditions.

##### pH Evaluation

The pH measurement of the formulated gels was measured using a digital pH meter (Mettler Toledo pH meter, USA). The pH measurements represent the mean ± standard deviation (SD) of three replicates.

##### Viscosity (Rheological Properties)

The rheological and flowability properties of the prepared gels were determined at room temperature using a TCV 300 viscometer (Cambridge applied laboratories viscometer, TX, USA). A piston of a range of 1–10 cP was used, as the formulations had a texture equivalent to water, and the temperature was set to room temperature (≈24 °C). One mL from each prepared hand sanitizer was filled into the measurement chamber. The chamber was capped for 60 s until it was stable, and then the data were recorded. The results represent the mean ± SD of three replicates.

##### Gel Spreadability

The spreadability of the prepared hand sanitizers was evaluated according to the methodology described in [18]; 0.5 gm of each formulated gel was spread on pre-marked transparent glass with a 2 cm diameter. Then, another transparent glass was placed on the top, followed by adding a 500 gm weight for 5 min to disperse the content. By this method, the spreadability was measured based on slip and drag characteristics of the gels. An excess of the gel was scrapped off from the edges. The diameter of the spreading area of each formulation was determined and represented by the mean ± SD of three replicates. The following equation was used to determine the spreading percentage:
(1)$$Spreadability\;\%=\frac{A2}{A1}\;\times \;100$$
where *A*1 is initial area before spreading (cm) and *A*2 is final area after spreading (cm).

#### 2.2.3. Microbial Suspension Preparation

Gram-positive and gram-negative bacteria, as well as an opportunistic pathogenic yeast (*C. albicans*), were obtained from the American Type Culture Collection (ATCC) as reference microbes to test the antimicrobial efficiency of prepared hand sanitizers. The bacterial isolates included: *Acinetobacter baumannii (A. baumannii)*—BAA 747, *Escherichia coli (E. coli)*—ATCC 25922, *K. pneumoniae*—BAA 1705, two strains of *Pseudomonas aeruginosa (P. aeruginosa)*—BAA 1744 and ATCC 27853, and *S. aureus* strains—ATCC 29,213 and BAA 977. Other bacterial strains were either isolated clinically or environmentally, which included: *E. coli*—isolates 1060, *Staphylococcus epidermidis (S. epidermidis)*—isolate 5029, *Staphylococcus hominis (S. hominis)*—isolate 5028, *Staphylococcus haemolyticus (S. haemolyticus)*—isolate 5034, and *Micrococcus luteus (M. luteus)*—isolate SB 115. The yeast *C. albicans*—ATCC 66,027 was obtained from ATCC. The bacterial and yeast suspensions, also known as inoculums, were all prepared in Mueller–Hinton broth by measuring 0.5 McFarland, according to [19]. All microorganisms were cultured on Mueller–Hinton agar medium and incubated at 37 °C overnight. 

#### 2.2.4. Antimicrobial Zone of Inhibition Test

To evaluate the antimicrobial activity of the prepared hand sanitizer gels, the zone of inhibition test against different gram-positive and gram-negative bacterial strains and a yeast was performed. Three commercially available hand sanitizers were also assessed as experimental controls. A final concentration of 1 × 10^6^ CFU/mL inoculum was equally distributed on the surface of agar plates. A sterile microbiological disc was dipped into each hand sanitizer gel, allowed to dry for a few seconds, and then positioned on the Mueller–Hinton agar plate. All plates were incubated overnight at 37 °C. The diameter of the clear area of no growth around each disc was recorded in millimeters (mm). The results represent the mean ± SD of three replicates.

#### 2.2.5. Skin Irritation Study (Acceptability Test)

Based on the results of the previous antimicrobial effectiveness test, the most efficient gel formulation was selected to be tested in a skin irritation study. The study was carried out on 20 volunteers and ethically approved by the research ethics committee in King Abdulaziz City for Science and Technology (KACST) (IRB approval number; IRB#20007). After explaining the research protocol with possible side effects, the volunteers were asked to sign consent forms. The assessment was performed by applying 1 mL of sanitizer gel on each volunteer’s palm, then allowed to stand for 5 min. 

A questionnaire was provided to all volunteers to conduct the acceptability test and skin irritation study. All volunteers had no clinical signs of dermal abrasion, trauma, or infection. The formulation was rated according to the characteristics of the hand sanitizer gel in terms of the product appearance, smell, texture, irritation or burning sensation, and redness after the application of the hand sanitizer gel. 

#### 2.2.6. Statistical Analysis

The results of the measured pH, viscosity, spreadability, and zone of inhibition were presented by the mean and SD of at least three replicates. The mean comparison of the antimicrobial activity of the prepared EO-based hand sanitizers and the commercially available ABHS were performed by T-test, and the *p*-value of < 0.05 was taken as a criterion for statistically significant difference. The statistical analysis was conducted using OriginPro 2016 software (OriginLab Corporation, MA, USA). 

## 3. Results and Discussion

### 3.1. Characterization and Evaluation of Hand Sanitizer Gels

#### 3.1.1. Organoleptic Test

The organoleptic test of hand sanitizer gels was conducted to evaluate the physical appearance of the prepared formulations. Following the visual quality inspection of the prepared hand sanitizer gels, the results indicated good characteristics observed for the tested formulations as follows. All gels were homogenous, clear with the EO’s distinctive odor, no syneresis occurred, the gels were easy to apply, light to spread, and had a consistent flow. There was a bubble-like appearance that was formed upon overnight storage, but disappeared after slight shaking. This was probably due to the addition of a considerably large volume of EOs (i.e., 1.25 or 2.5% *v*/*v*). The hand sanitizer gels exhibited no coarse particles upon spreading on a transparent glass, owing to the homogeneity of the prepared formulations. All of the observed results were consistent with other hand gels of previous studies [20,21].

#### 3.1.2. pH Evaluation

The pH values of the formulated hand sanitizer gels were measured using a digital pH meter. The study was conducted to check the neutralization of different prepared formulations. The ideal standards for a pH value of a topical dosage form should be within the broad pH range of the skin, i.e., 4.0 to 7.0, in order to avoid skin inflammation and irritation [22]. The pH measurements in Table 2 showed that all prepared formulations were slightly acidic, with pH values around 3.9. This might be due to the large proportion of aloe vera (90% *v*/*v*) with a natural acidic pH (4.0–4.5). 

It was reported that the optimal condition for the growth of several pathogenic bacteria that can infect the skin is in the neutral pH range. In contrast, normal flora is more likely to be settled in the skin if the pH condition is slightly acidic [23]. It should be noted that the acidic pH environment (4.0–4.5) can enhance the attachment of the normal flora to the skin [22]. Moreover, it was demonstrated that the average pH value of the skin surface’s natural condition is below 5.0, which is also an optimum condition for several necessary dermal biological processes, such as the homeostasis of the stratum corneum formation of the lipid barrier [24,25]. The acidic condition can also enhance the activity of the antimicrobial compounds against pathogenic microorganisms. For instance, the antimicrobial peptide Dermcidin found in the sweat showed higher antimicrobial activity against *S. aureus* at a pH of 5.5 compared to a pH of 6.5 [26]. Furthermore, it was reported from a long-term clinical trial study that the growth of *Propionibacteria* after washing the forehead and forearm with neutral synthetic formulation was higher than that observed when a slightly acidic formulation was applied [27]. Therefore, the slightly acidic gel formulations could be advantageous in the antimicrobial applications, as they can be more effective against pathogenic microbial growth.

#### 3.1.3. Viscosity (Rheological Properties)

The viscosity of the prepared gel formulations is one of the fundamental parameters that should be controlled, as it can reflect the consistency and flowability of the gel formulations when applied to the skin [28]. In this study, the viscosity test was carried out to determine the thickness of the preparations using a TCV 300 viscometer and to explore the influence of gel components on the products’ rheological properties. As shown in Table 3, all prepared formulations’ viscosities were higher than the viscosity of water and ethanol (0.9 cP). The slightly higher viscosity values among all prepared hand sanitizer gels (>0.9 cP) were probably due to the use of EOs, as the EO-free gel demonstrated an equivalent viscosity to water and ethanol (0.9 cP). 

#### 3.1.4. Gel Spreadability

The spreadability plays a critical role in the application of hand sanitizers, and is associated with consumer compliance and uniformity of the applied gels to meet topical application quality standards. Hence, the gel spreadability test was carried out to assess the ability of the prepared hand gels to distribute properly when applied to the skin, in which the optimal gel formulation should have less spreading time (i.e., high spreadability). One of the main parameters that can affect the gel spreadability is the viscosity of the formulation, in which a lower viscous gel has higher spreadability. 

Table 4 shows the spreadability values of all prepared gels, which are found in the range of 558 to 638%. Maximum spreadability was observed for gels containing 1.25% (*v*/*v*) clove oil (F2), followed by 2.5% (*v*/*v*) lavender oil (F3) and 2.5% (*v*/*v*) tea tree oil (F4), whereas lower spreadability was measured for 2.5% (*v*/*v*) clove oil (F1). The results also conformed to the viscosity study, as an increase in the viscosity value of 2.5% (*v*/*v*) clove oil hand sanitizer (1.1 cP) showed a decrease in the spreadability (558%) of this gel. 

### 3.2. Antimicrobial Zone of Inhibition Assay

The zone of inhibition test was conducted to evaluate the antimicrobial efficacy of the prepared hand sanitizers compared to three commercially available hand sanitizers (C1, C2, and C3) against 12 gram-negative and gram-positive bacterial strains, as well as *C. albicans* yeast. Well-defined zones of inhibition were observed with variable diameters, as shown in (Figure 1 and Figure 2). 

The zone of inhibition diameters of the prepared hand sanitizers demonstrated that both concentrations used in the clove oil hand sanitizers (1.25% and 2.5% *v*/*v*) inhibited all bacterial strains and the yeast (13 microorganisms in total). In contrast, the tea tree hand sanitizer had the highest efficient antibacterial activity (inhibiting the 12 bacterial strains more efficiently), but lacked anti-yeast activity. The lavender hand sanitizer was the least efficient hand sanitizer gel, with the least antibacterial inhibition spectrum (effective against 10 bacterial strains), and had no anti-yeast activity. All commercially available hand sanitizers showed antibacterial efficiencies against all bacterial strains, but with variable competences, owing to their alcohol-based nature, which is known for its effective antibacterial activity against several gram-positive and gram-negative bacterial strains [29]. Interestingly, only the ethanol-based commercial hand sanitizer gel (C1) and the clove oil hand sanitizers (F1 and F2) showed anti-yeast activities. The zone of inhibition diameters of all tested hand sanitizers against gram-negative and gram-positive bacteria, as well as *C. albicans*, are presented in Table 5, Table 6 and Table 7, respectively.

The antimicrobial activity of the prepared EO-based hand sanitizers and the commercially available ABHS were compared. It was shown that the antimicrobial effectiveness of C1 was significantly (*p < 0.05*) higher than all EO-based hand sanitizers, except for the tea tree formulation (F4), which showed an insignificant difference. Additionally, the antibacterial activity of F4 was significantly (*p < 0.05*) higher than the other ABHS (C2 and C3). Both clove oil (F1 and F2) and lavender (F3) hand sanitizers demonstrated insignificant antimicrobial activities with C2 and C3. The anti-yeast efficacy of hand sanitizers can be considered as an important feature of hand sanitizers, and it was lacking in the tea tree hand sanitizer, despite its remarkable antibacterial activity.

The EO-free control formula showed no antibacterial effect against all tested microorganisms, except *S. epidermidis*. The lack of antibacterial effectiveness of aloe vera is inconsistent with [30] and [31], which reported antibacterial activity against different bacterial strains that include *E. coli, Proteus vulgaris, S. aureus, S. epidermidis,* and *Streptococcus pyogenes (S. pyogenes*). This indicated that the EO could enhance the antimicrobial effect of the prepared hand sanitizers. However, in this study, the aloe vera gel was used for its natural moisturizing ability through the humectant mechanism, as reported in [32], and to counteract the burning sensation of EOs through its wound healing ability, as reported in [13]. The variability of the antimicrobial activity of each EO should be considered, as not all EOs have similar individual compliance. For instance, the rate of acceptance of clove oil (2.5% *v*/*v*) was low due to its strong smell and skin burning sensation, as will be described in the next section (3.3). Therefore, half of the initially used concentration (i.e., 2.5% *v*/*v*) of clove oil, equivalent to 1.25% *v*/*v* (F2), was used. The antimicrobial activity of this formulation demonstrated a similar antimicrobial spectrum against all test bacterial and yeast strains to the 2.5% (*v*/*v*) clove oil hand sanitizer (F1), but with slightly lower effectiveness, which was expected. 

Overall, the antimicrobial assessment exhibited that 2.5% (*v*/*v*) clove oil hand sanitizer can inhibit bacterial and yeast microorganisms. It was previously reported that clove oil has an antifungal effect against different dermatophytic fungi, when used at a concentration range of 1 to 5% (*v*/*v*) [33]. Nzeako et al. also reported the antibacterial activity of clove oil against other microbes, including *S. aureus, P. aeruginosa, E. coli, S. pyogenes, Corynebacterium species, Salmonella species, Bacteroides fragilis,* and *C. albicans* [34]. Therefore, clove oil hand sanitizers (F1 and F2) were selected for the acceptability study to assess the safety of these hand sanitizers upon application. 

### 3.3. Acceptability Test (Skin Irritation Study)

The skin irritation study was performed on 20 volunteers, and the results are presented in the (Appendix A). According to the pH evaluation results, viscosity, spreadability, and antimicrobial activity, the gel formulation containing 2.5% (*v*/*v*) of clove oil (F1) was selected for the acceptability test and skin irritation study. Ideal hand sanitizers should possess a pleasant smell, feel comfortable upon use, be easy to apply and not sticky, and have an excellent antimicrobial activity. 

The skin irritation study results showed that the hand sanitizer gel containing 2.5% (*v*/*v*) clove oil was very well-tolerated, and did not produce any sign of irritation or skin redness after being applied to the participants. However, a minimal sense of itching was reported in five volunteers out of 20, four of whom already suffered from a skin condition, namely eczema, and demonstrated redness. Therefore, 1.25% (*v*/*v*) clove oil hand sanitizer gel (F2) was applied again to those four volunteers, and no side effects were reported.

## 4. Conclusions

Hand sanitizer gel is one of the alternative options for hand hygiene. Due to the emergence of the COVID-19 pandemic, the prevention and control of bacterial or fungal co-infections using AFHS gels can be crucial, particularly when the alcohol supply chain is at risk. In this study, AFHS gels were formulated using aloe vera, glycerin, vitamin E, and several EOs as the active antimicrobial ingredients. It is concluded from the results that the prepared formulations have excellent organoleptic properties, pH values comparable to skin pH, and suitable viscosity and spreadability profiles. 

The antimicrobial test showed varying activities of different EO-based formulations against several gram-positive and gram-negative bacteria and Candida. The results provided evidence that clove oil exhibited a profound antimicrobial activity against a broad range of microbes. The widest antimicrobial spectrum was observed with 2.5% (*v*/*v*) clove oil hand sanitizer (F1), which showed an antimicrobial activity close to the experimental ABHS controls. However, slight skin irritation sensation was observed in 20% of the volunteers. Instead, 1.25% (*v*/*v*) clove oil hand sanitizer (F2) was incorporated in the gel to prepare a superior antimicrobial product with slight or no adverse effects and higher acceptability for human skin. However, more research should be directed in future prospects to assess the efficacy against more bacterial species, yeast, and fungus. Furthermore, the antiviral activity of clove oil hand sanitizer should also be assessed to confirm its antiviral effectiveness, in order to be used as a potential and more effective alternative to ABHS during pandemics. Finally, the stability test for the hand sanitizer formulations should also be evaluated to ensure the shelf-life of this EO-based hand sanitizer.

## Figures and Tables

**Figure 1 ijerph-18-06252-f001:**
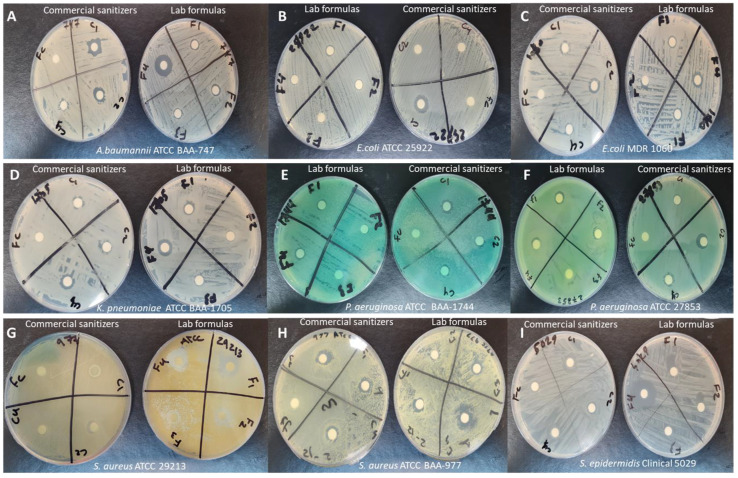
The zone of inhibition diameters of the prepared hand sanitizers compared to three commercially available hand sanitizers (C1, C2, and C3) against (**A**): *A. baumannii*—BAA 747; (**B**): *E. coli*—ATCC 25922; (**C**): *E. coli*—MDR 1060; (**D**): *K. pneumoniae*—BAA 1705; (**E**): *P. aeruginosa*—BAA 1744; (**F**): *P. aeruginosa*—ATCC 27853; (**G**): *S. aureus*—ATCC 29213; (**H**): *S. aureus*—BAA 977; (**I**): *S. epidermidis*—isolate 5029; (**J**): *S. homini*—isolate 5028; (**K**): *S. haemolyticus*—isolate 5034; (**L**): *M. luteus*—isolate SB 115; and (**M**): *C. albicans*—ATCC 66027. The clove EO-based hand sanitizers in both concentrations inhibited all bacterial strains and the yeast (13 microorganisms in total). The tea tree hand sanitizer had the highest efficient antibacterial activity (inhibited the 12 bacterial strains), but lacked anti-yeast activity. The lavender hand sanitizer was the least efficient hand sanitizer gel, with the least antibacterial inhibition spectrum (effective against 10 bacterial strains) and no anti-yeast activity. The commercially available hand sanitizers were able to inhibit all bacterial strains, but with variable efficacies, while the oil-free control formula showed no antibacterial effect against all tested bacteria, except *S. epidermidis*. The ethanol-based control (C1) showed anti-yeast activity against *C. albicans*.

**Figure 2 ijerph-18-06252-f002:**
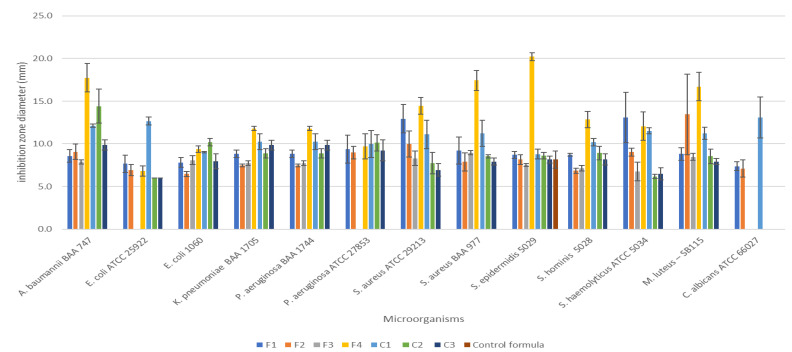
The zone of inhibition diameters of the prepared hand sanitizers compared to three commercially available hand sanitizers (C1, C2, and C3) against the 12 gram-negative and gram-positive bacterial strains and *C. albicans* yeast. The results represent the mean of three independent experiments ± SD (n = 3). It was shown that the clove hand sanitizers (F1 and F2) were able to inhibit all bacterial strains and the yeast, similar to one control (C1). In contrast, the tea tree hand sanitizer had the highest efficient antibacterial activity, but lacked anti-yeast activity.

**Table 1 ijerph-18-06252-t001:** The composition of prepared hand sanitizer gels.

Formulation Number	Essential Oils	Aloe Vera	Glycerin	Vitamin E	Distilled Water
F1	2.5% (*v*/*v*)—Clove oil	90% (*v*/*v*)	5% (*v*/*v*)	0.05% (*v*/*v*)	2.45% (*v*/*v*)
F2	1.25% (*v*/*v*)—Clove oil				3.70% (*v*/*v*)
F3	2.5% (*v*/*v*)—Lavender oil				2.45% (*v*/*v*)
F4	2.5% (*v*/*v*)—Tea tree oil				2.45% (*v*/*v*)
Control	None				4.95% (*v*/*v*)

**Table 2 ijerph-18-06252-t002:** The pH values of hand sanitizer gel formulations. The result represents the mean ± SD of three replicates (*n* = 3).

Formulation	pH
F1	3.9 ± 0.0
F2	3.9 ± 0.0
F3	3.9 ± 0.1
F4	3.9 ± 0.1
Control	3.9 ± 0.0

**Table 3 ijerph-18-06252-t003:** The viscosity values of hand sanitizer gel formulations. The result represents the mean ± SD of three replicates (*n* = 3).

Formulation	Viscosity (cP)
F1	1.1 ± 0.10
F2	1.0 ± 0.02
F3	1.0 ± 0.03
F4	1.0 ± 0.02
Control	0.9 ± 0.01
Water	0.9 ± 0
Ethanol	0.9 ± 0

**Table 4 ijerph-18-06252-t004:** The spreadability values of hand sanitizer gel formulations. The result represents the mean ± SD of three replicates (*n* = 3).

Formulation	Spreadability (%)
F1	558 ± 3
F2	638 ± 3
F3	622 ± 6
F4	622 ± 3

**Table 5 ijerph-18-06252-t005:** The zone of inhibition diameters of the prepared hand sanitizers compared to three commercially available hand sanitizers (C1, C2, and C3) against gram-negative bacterial strains. The results represent the mean of three independent experiments ± SD (*n* = 3). Both the clove and tea tree EO hand sanitizers showed antibacterial activities against all gram-negative strains similarly to the controls, but with variable efficacies. The oil-free control formula showed no antibacterial effect against any of the tested gram-negative bacteria.

Formulation	*A. baumannii* BAA 747 (mm)	*E. coli* ATCC 25,922 (mm)	*E. coli* 1060 (mm)	*K. pneumoniae* BAA 1705 (mm)	*P. aeruginosa* BAA 1744 (mm)	*P. aeruginosa* ATCC 27,853 (mm)
F1	9 ± 1	8 ± 1	8 ± 1	9 ± 0	9 ± 0	9 ± 2
F2	9 ± 1	7 ± 1	7 ± 0	8 ± 0	8 ± 0	9 ± 1
F3	8 ± 0	0	8 ± 1	8 ± 0	8 ± 0	0
F4	18 ± 2	7 ± 1	9 ± 0	12 ± 0	12 ± 0	10 ± 2
C1	12 ± 0	13 ± 1	9 ± 0	10 ± 1	10 ± 1	10 ± 2
C2	14 ± 2	6 ± 0	10 ± 0	9 ± 1	9 ± 1	10 ± 1
C3	10 ± 1	6 ± 0	8 ± 1	10 ± 1	10 ± 1	9 ± 1
Control	0	0	0	0	0	0

**Table 6 ijerph-18-06252-t006:** The zone of inhibition diameters of the prepared hand sanitizers compared to three commercially available hand sanitizers (C1, C2, and C3) against gram-positive bacterial strains. The results represent the mean of three independent experiments ± SD (*n* = 3). All EO hand sanitizers showed antibacterial activities against all gram-positive strains similarly to the controls, but with variable efficacies. The oil-free control formula showed no antibacterial effect against any of the tested gram-positive bacteria, except *S. epidermidis*.

Formulation	*S. aureus* ATCC 29,213 (mm)	*S. aureus BAA* 977 (mm)	*S. epidermidis* 5029 (mm)	*S. hominis* 5028 (mm)	*S. haemolyticus* ATCC 5034 (mm)	*M. luteus* SB115 (mm)
F1	13 ± 2	9 ± 2	9 ± 0	9 ± 0	13 ± 3	9 ± 1
F2	10 ± 2	8 ± 1	8 ± 1	7 ± 0	9 ± 0	14 ± 5
F3	8 ± 1	9 ± 0	8 ± 0	7 ± 0	7 ± 1	9 ± 0
F4	14 ± 1	17 ± 1	20 ± 1	13 ± 1	12 ± 2	17 ± 2
C1	11 ± 5	11 ± 2	9 ± 1	10 ± 0	12 ± 0	11 ± 1
C2	8 ± 1	9 ± 0	9 ± 0	9 ± 1	6 ± 0	9 ± 1
C3	7 ± 1	8 ± 0	8 ± 0	8 ± 1	7 ± 1	8 ± 0
Control	0	0	8 ± 1	0	0	0

**Table 7 ijerph-18-06252-t007:** The zone of inhibition diameters of the prepared hand sanitizers compared to three commercially available hand sanitizers (C1, C2, and C3) against *C. albicans*. The results represent the mean of three independent experiments ± SD (*n* = 3). Only the clove hand sanitizers (F1 and F2) and the ethanol-based control (C1) showed anti-yeast activity against *C. albicans*, but with variable efficacies.

Formulation	*C. albicans* ATCC 66,027 (mm)
F1	7 ± 1
F2	7 ± 1
F3	0.00
F4	0.00
C1	13 ± 2
C2	0.00
C3	0.00
Control	0.00

## Data Availability

The data presented in this study are available on request from the corresponding author.

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
