# Peer review of "Formulation and Evaluation of Alcohol-Free Hand Sanitizer Gels to Prevent the Spread of Infections during Pandemics"

_ijerph, 2021, doi:10.3390/ijerph18126252_

Round 1

Reviewer 1 Report

The authors didn't directly answer my first question. Which ingredients are essential for the antimicrobial activity of this sanitizer? For the second point, considering the limitation to perform live virus infection experiments, the description about antiviral activity of this sanitizer should be deleted or properly described in this manuscript.

Author Response

We thank the referee for their thoughtful and helpful comments.  We respond to each point below; their suggestions have undoubtedly improved the paper, for which we are very grateful.  

Point 1: The authors didn't directly answer my first question. Which ingredients are essential for the antimicrobial activity of this sanitizer?

Response 1: Thank you for this comment. The primary active compound of the alcohol-free hand sanitizer is essential oils (EOs), which have a wide range of antimicrobial activities. The different AFHS formulations used in this work have the same concentration of additives (aloe vera, glycerin and vit. E - Table.1), whereas the only variable in the prepared formulations is the EOs. The EO-free control formula showed no antibacterial effect against all tested microorganisms, except S. epidermidis. This indicated that the EO possesses an antimicrobial effect, hence it is the active compound.

We improved the introduction section by adding the following statements

‘The primary active compounds of the AFHS gels are EOs, which have a wide range of antimicrobial activities [11]. The antimicrobial activity of EOs reported to be due to their hydrophobicity nature that facilitates the partition of active components in the lipid of bacterial cell membrane and mitochondria, hence, reduce the cytoplasmic membrane integrity [12].’

This statement can be found in the intro section – lines 79 -83.

Point 2: For the second point, considering the limitation to perform live virus infection experiments, the description about antiviral activity of this sanitizer should be deleted or properly described in this manuscript.

Response 2: The description of antiviral activity of the prepared AFHS was deleted from the revised manuscript.

Additional English editing was performed to the manuscript. The yellow highlighted sections were majorly amended/newly added.

Reviewer 2 Report

Thank you for great support contributed to the revision, the manuscript look more professional and sound scientific paper. It may need some polishing work in English and presentation.

Best of luck

Author Response

We thank the Referee for his/her valuable time for reviewing and kindly giving us constructive suggestions to improve our manuscript. Additional English editing was performed to the manuscript as requested. The yellow highlighted sections were majorly amended/newly added.

Reviewer 3 Report

Why didn't you run an analysis to determine the statistical difference between all groups based on the inclusion of essential oils and commercial hand sanitizer?

Add units in tables.

Author Response

We thank the referee for their thoughtful and helpful comments.  We respond to each point below; their suggestions have undoubtedly improved the paper, for which we are very grateful.  

Point 1: Why didn't you run an analysis to determine the statistical difference between all groups based on the inclusion of essential oils and commercial hand sanitizer?

Response 1: Thank you for this valuable comment, we considered performing a statistical analysis comparing the general antimicrobial activity of the prepared hand sanitizers against the commercially available ones.

In the methodology section 2.2.6., the following paragraph has been improved

‘The results of the measured pH, viscosity, spreadability, and zone of inhibition were presented by the mean and SD of at least three replicates. The mean comparison of the antimicrobial activity of the prepared EO-based hand sanitizers and the commercially available ABHS was performed by T-test and the p value of < 0.05 was taken as a criterion for a statistically significant difference. The statistical analysis was conducted using OriginPro 2016 software (OriginLab Corporation, Massachusetts, USA)’ [Lines 199-204]

In the results section 3.2, the following paragraph has been added

The antimicrobial activity of the prepared EO-based hand sanitizers and the commercially available ABHSs were compared. It was shown that the antimicrobial effectiveness of C1 is significantly (p < 0.05) higher than all EO-based hand sanitizers, except for the tea tree formulation (F4), which showed an insignificant difference. Additionally, the antibacterial activity of F4 was significantly (p < 0.05) higher than the other ABHS (C2 and C3). Both clove oil (F1 and F2) and lavender (F3) based hand sanitizers demonstrated insignificant antimicrobial activities with C2 and C3. The anti-yeast efficacy of hand sanitizers can be considered as an important feature of hand sanitizers, in which it lacked in the tea tree-based hand sanitizer in spite of its remarkable antibacterial activity.’ [Lines 301-309]

Point 2: Add units in the tables.

Response 2: This has been resolve by adding the unit in brackets for tables 3-7 (above each numerical-containing column), except for Table 2 which does not require. In Table 1, the units are beside each value.

Additional English editing was performed to the manuscript. The yellow highlighted sections were majorly amended/newly added.

Round 2

Reviewer 1 Report

All of my concerns have been addressed and the manuscript reads well. I suggest to publish this work in International Journal of Environmental Research and Public Health.

This manuscript is a resubmission of an earlier submission. The following is a list of the peer review reports and author responses from that submission.

Round 1

Reviewer 1 Report

Dear Authors,

Thank you for a great effort on this interesting manuscripts, it would be of benefit to the societies.

  1. You may need some grammar editing to make your work more professional and succinct; for example line 42-68, it's quite clumsy with repeated words where you can use compound or complex sentence instead of multiple short simple sentences. Another point, figure more than 10 should be present as number such as 70 million in line 37 and 20 volunteers in line 184. Please go through your manuscript and make editing as appropriate.
  2.  You may cut down the background (from 5 lines to 2 lines) and add more results as the results is the most important section of the  abstract and your team worked hard on exploring those great outcomes . The audiences would like to learn new knowledge from your study.
  3. The details of '2.2.1. Preparation of Hand Sanitizer Gels' is not flow, please revise the paragraph. Also, which method you use in preparing gel formula, it 'd be better to present clearly.
  4. You have run many tests for the gel formula, impressive. Do you test the stability of gel?? it's important quality of gel product
  5. Please explain more in '2.2.6. Statistical Analysis'. Whare were variables been measured and tested. How the categorical variables been measured, presented and tested  such as %, frequency because you cannot present mean and SD for categorical data. Please revise and add more details in  2.2.6 

Reviewer 2 Report

Article title: Formulation and Evaluation of Alcohol-Free Hand Sanitizer Gels to Prevent the Spread of Infection during Pandemics.

Comments:

Major comments:

  1. There is lack of novelty of the work and not suitable for publishing in journal like International Journal of Environmental Research and Public Health.

Reason: Alcohol-based hand sanitizers were found more effective than alcohol-free hand sanitizers against different bacteria and viruses including SARS-CoV-2. There are so many studies where alcohol-based hand sanitizers were found effective with good dermal tolerability*. So in a pandemic situation, how this alcohol free hand sanitizer formulation will be effective against SARS-CoV-2? There is no efficacy test against SARS-CoV-2 or other viruses in this study. So in a pandemic situation, if the prepared hand sanitizers are not effective against the pandemic-causing virus, what is the benefit of this formulation over alcohol-based hand sanitizers?

On the other hand, works are ongoing that, how alcohol free hand sanitizers can be effective against SARS-CoV-2.**

*Annika K, Daniel T, Philip Vk, Silvio S, Mitra G, Tran Thi Nhu T, et al. Inactivation of Severe Acute Respiratory Syndrome Coronavirus 2 by WHO-Recommended Hand Rub Formulations and Alcohols. Emerging Infectious Disease journal. 2020;26(7).

*Kenters N, Eikelenboom-Boskamp A, Hines J, McGeer A, Huijskens EGW, Voss A. Product dose considerations for real-world hand sanitiser efficacy. Am J Infect Control. 2020;48(5):503-6.

**https://www.eurekalert.org/pub_releases/2020-12/byu-ahs120120.php

Again, there are so many essential oil based hand sanitizers available in the market. The authors have mentioned that there are shortage of alcohol based hand sanitizers, but if there are stocks of alcohol free hand sanitizer in the market, people can buy those. They have compared their formulation with marketed alcohol based hand sanitizers; they also need to compare their prepared hand sanitizer with marketed alcohol free hand sanitizer to show how effective your formulation is.

  1. This manuscript overlooked so many issues which needs to be incorporated and there is lack of scientific writings like information gap, proper scientific way of presenting data.

Some examples are-

  1. They have suggested that, people can prepare this hand sanitizer at home. They need to mention the precautions that people need to be taken while preparing this hand sanitizer. This is a gel based hand sanitizer and not a solution, how general people can ensure the homogeneity in mixing at home? Without having precautionary section or specific guideline, it is not proper way of writing that people can prepare such formulation at home.
  2. They have not mentioned the sources and purity of the materials (Aloe vera, glycerine, vitamin E, and essential oils) which are very important. According to the manuscript, it can be prepared at home, so without purity and necessary information how it can be fruitful?
  3. They have to mention the amount of vitamin E and other chemicals in mL rather than drop. The amount will be varied in terms of droppers.

Minor comments:

  • If there is no efficacy study against the pathogens causing pandemic like SARS-CoV-2, then it should be mentioned, how the stated bacteria/yeast (study they have conducted on different bacteria and yeast) could worsen the pandemic situation as they have mentioned the terms pandemic in the title and the introduction they have mentioned about COVID-19. Therefore, there should have a brief description of this in the introduction.
  • There are different types of hand sanitizer formulation like gel, solution, or foam. Why do they have chosen gel formulation over solution or foam formulation? What are the advantages and disadvantages of this gel form? Thy have to mention it in the introduction.
  • They have to state the advantages as well as the limitations of natural components. Natural components are not readily available all the time as they are seasonal and costly. On the other hand, their purity varied significantly. Many people are allergic to natural compounds, so need to mention these issues in the introduction.
  • Organoleptic test. Is it shape or texture? (Page 3, Line 132)
  • 5 out of 20 volunteers faced skin irritations meaning 25% of patients had faced skin irritations which is a big number. On the other side, if 4 out of these 5 volunteers have previous skin diseases then how they were included in the study?
  • In which form all the commercial hand sanitizers were? Either all in gel, foam, or solution? Again, no detailed information was provided about the commercial hand sanitizers like what was the percentage of alcohol content? If the alcohol content is less than 60% (which is recommended), that will not be effective and should not be included in the study.
  • The commercial sample C1 was found more effective (inhibition zone diameter, Figure 1) compared to the clove 2.5% formulation (which was optimized later for acceptability test). Then what is the novelty of the work if it is not more effective than the alcohol-based hand sanitizer? Some formulations they have studied containing other components like tea tree extracts showed some good anti-bacterial efficiency so combinations of clove and tea tree can be a good choice of components.

Reviewer 3 Report

The authors report e a home-based alcohol-free hand sanitizer with antimicrobial properties. The antimicrobial activity was evaluated in several Gram-positive and gram-negative bacteria as well as yeast. The overall finding is potential interesting and the results look convincing. However, even if the author mentioned the sensitizer should be able to inactive the virions, there is not supporting data showing the antiviral activity.

  1. The authors mentioned that several virus including HCV, HSV, etc can be inhibited clove active ingredients. But whether antimicrobial activity of this sanitizer is due to one of the ingredients was not investigated.
  2. The antiviral activity of the sanitizer should be investigated. It will be not easy to test SARS-CoV-2 which requires BSL-3 Labs. But the authors should test one or several representatives from other families.